# Whole-Body Soft-Tissue Lesion Tracking and Segmentation in Longitudinal CT Imaging Studies

**Alessa Hering**[*1,2]                    ALESSA.HERING@MEVIS.FRAUNHOFER.DE
[1] *Fraunhofer MEVIS, Bremen, Germany*
[2] *Diagnostic Image Analysis Group, Radboudumc, Nijmegen, Netherlands*

**Felix Peisen**[*3]                    FELIX.PEISEN@MED.UNI-TUEBINGEN.DE
[3]*Department of Diagnostic and Interventional Radiology, University Hospital Tübingen, Germany*

**Teresa Amaral**[4]                    TERESA.AMARAL@MED.UNI-TUEBINGEN.DE
[4]*Department of Dermatology, University Hospital Tübingen, Germany*

**Sergios Gatidis**[3]                    SERGIOS.GATIDIS@MED.UNI-TUEBINGEN.DE

**Thomas Eigentler**[4]                    THOMAS.EIGENTLER@MED.UNI-TUEBINGEN.DE

**Ahmed Othman**[3]                    AHMED.OTHMAN@MED.UNI-TUEBINGEN.DE

**Jan Moltz**[1]                    JAN.MOLTZ@MEVIS.FRAUNHOFER.DE

**Editors:** Under Review for MIDL 2021

## Abstract

In follow-up CT examinations of cancer patients, therapy success is evaluated by estimating the change in tumor size. This process is time-consuming and error-prone. We present a pipeline that automates the segmentation and measurement of matching lesions, given a point annotation in the baseline lesion. First, a region around the point annotation is extracted, in which a deep-learning-based segmentation of the lesion is performed. Afterward, a registration algorithm finds the corresponding image region in the follow-up scan and the convolutional neural network segments lesions inside this region. In the final step, the corresponding lesion is selected. We evaluate our pipeline on clinical follow-up data comprising 125 soft-tissue lesions from 43 patients with metastatic melanoma. Our pipeline succeeded for 96% of the baseline and 80% of the follow-up lesions, showing that we have laid the foundation for an efficient quantitative follow-up assessment in clinical routine.

**Keywords:** Soft-tissue lesion, follow-up, CT, Lesion Tracking, Lesion Segmentation, Image Registration

## 1. Introduction

Measurement of metastatic tumors on longitudinal computer tomography (CT) scans is essential to evaluate the efficacy of cancer treatment. The current guideline of metastatic tumor evaluation on CT scans is called response evaluation criteria in solid tumors (RE-CIST) (Eisenhauer et al., 2009). Manual measurement of the tumors for the RECIST criteria is often time-consuming and error-prone. However, the diameter-based RECIST criteria also undergo continuous changes. Automated approaches might significantly speed up response evaluation and help to handle the ever-growing mass of image-based staging and follow-up evaluations (Moawad et al., 2020).

---

[*] Contributed equally

Furthermore, radiomics is currently one of the most important topics in radiology. High-throughput extraction of quantitative features resulting in the conversion of medical images into minable data and the subsequent analysis promise new insights into therapy response and hold the potential to revolutionize medical image-based evaluation techniques (Gillies et al., 2016). Both fields have a huge clinical impact due to rising demand for fast and reliable therapy response evaluations. They, however, share a common bottleneck: automated lesion segmentation. Only if this obstacle is overcome, clinicians will use the mentioned techniques accordingly in a daily manner.

Metastatic malignant melanoma is the perfect entity to implement a pipeline for full-body lesion segmentation. Besides lung and liver, metastatic lesions of melanoma can be found in almost every organ or tissue, such as lymph nodes, adrenal glands, cerebrum, bone, spleen, and soft tissue (Schadendorf et al., 2018). Whole-body cross-sectional imaging is part of the standard diagnostic work-up for staging, response assessment, and follow-up in patients with advanced melanoma according to current international guidelines. Malignant melanoma has been increasing fast in the last decades and represents a public health matter in several countries due to its high mortality rates (Ward and Farma, 2017).

Among melanoma metastases, soft-tissue lesions provide a particular hurdle. They can arise in a variety of locations (cutaneous, subcutaneous, muscular, retroperitoneal) and shapes (round, multilobular, well defined, invasive), are often primarily small and, if not surrounded by fatty tissue, extremely hard to distinguish. A sufficient segmentation pipeline for soft-tissue metastases in malignant melanoma patients would therefore provide a valuable foundation for further steps towards a full-body lesion segmentation pipeline, that could be transferred to other entities.

To the best of our knowledge, no work has been presented until now that tackles the problem of soft-tissue lesion segmentation in longitudinal CT image series. Lesion segmentation in other anatomical regions, however, has been studied extensively. For example, promising results have been accomplished for liver (Bilic et al., 2019) and kidney lesions (Heller et al., 2021) in challenges. Currently, the most general and successful avilable approach is the nnU-Net framework of (Isensee et al., 2020), which has shown impressive results for several organ segmentation tasks such as liver, spleen, kidney, pancreas, heart, or aorta segmentation and also outperforms most methods segmenting different lesion types such as pancreas, liver, lung, kidney, or MS lesions. nnU-Net (Isensee et al., 2020), initially based on U-Net (Ronneberger et al., 2015), automatically configures itself, including preprocessing, network architecture, training and post-processing—making it an ideal baseline to build a lesion tracking pipeline.

However, as the lesion segmentation experiments in (Isensee et al., 2020) focus only on segmenting lesions in one organ in one scan, it cannot be used "as is" and requires some modifications. Only few works have been presented on lesion tracking (Cai et al., 2020) and on lesion tracking and segmentation in longitudinal image scans (e.g. (Xu et al., 2011; Moltz et al., 2012; Folio et al., 2013). In this work, we tackle the problem of longitudinal tracking and segmentation of soft-tissue lesions in whole-body CT scans.

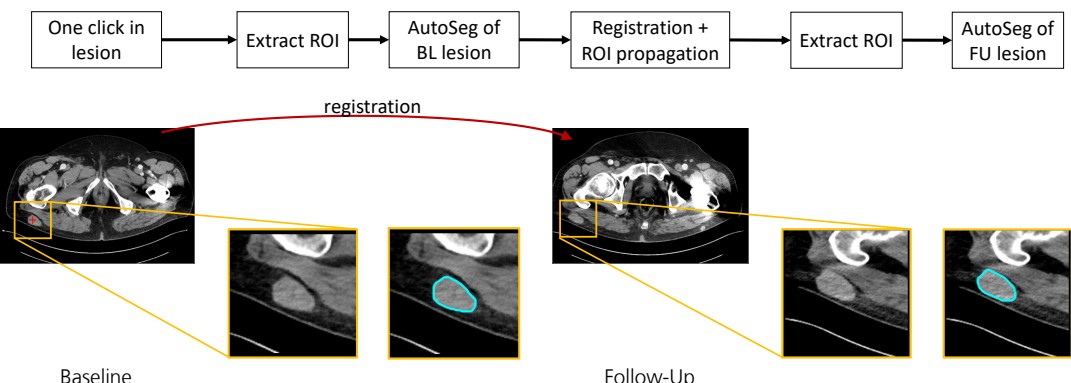

Figure 1: Schematic representation of the proposed pipeline for lesion tracking and segmentation.

## 2. Method

In our proposed pipeline, soft-tissue lesions are first identified by a radiologist with one click inside the lesion in the baseline CT scan. This step is introduced to avoid annotation of false positive lesions. We then apply our algorithm to automatically segment and measure the diameter in the baseline and follow-up image. This is done by (1) extracting the region of interest (ROI) around the point annotation of the radiologist and applying our CNN to segment the lesion; (2) registering the baseline to the follow-up image; (3) propagating the region of interest to the follow-up image to constrain the search region and applying the CNN on the propagated region of interest in the follow-up image; and (4) selecting the corresponding lesion in the output of the CNN. Figure 1 shows an overview of our proposed algorithm. In the following, we describe each step in more detail.

### 2.1. Lesion Segmentation

To generate the training data, we select for each lesion a bounding box around the point annotation of the radiologist with a size of 100 mm, which is clamped by the image region. Then, we use the nnU-Net framework of (Isensee et al., 2020) to train a 3d full resolution model which consists of a U-Net-like (Ronneberger et al., 2015) architecture. The main settings are shown in Table 1 in the appendix. The trained network is applied to segment the lesion in the baseline and follow-up image on the test dataset.

### 2.2. Registration

Propagation of lesion segmentations into follow-up images of the same patient allows for a higher degree of automation because the location and approximate appearance of the lesions are already known. In this scenario, registration algorithms can be employed to find the corresponding image region (Moltz et al., 2012). For metastatic melanoma, typically full-body or thorax-abdomen CT scans are acquired, which can easily exceed image sizes of $512 \times 512 \times 1000$, which can be a challenge in terms of memory usage and runtime.

The registration has to align the global structures but at the same time be locally accurate enough so that the lesion propagation is precise enough. Therefore, we adopted a three-step approach to automatically register the baseline to the follow-up image, which consists of the following steps: (1) a translational alignment; (2) a rigid registration; and (3) a deformable registration. Hereby, the registration pipeline starts with robust methods with fewer degrees of freedom and moves on to more precise, but less robust methods, which require better starting points due to their higher degrees of freedom.

**(1) Translational Alignment**    The translational prealigment is based on a brute force grid search method named FASTA (Fast Translation Alignment), which evaluates a difference measure (here Sum-of-Squared-Distances (SSD), the squared $\ell_2$ norm of the difference image) on a grid of possible translations. Finer grids allow for more precise translation estimation at the expense of increased computational cost. For faster processing, the moving image is resampled to a maximal image size of $128 \times 128 \times 128$. The fixed image is resampled to the same image resolution as the moving image. For the grid generation, we choose a sampling rate of 3, 3, and 51 in x, y, and z-direction respectively. Since the CT scans are centered around the body center, only the z-translation is used for prealignment.

**(2) Rigid Registration**    The translational prealignment in z-direction is used as a starting point for a rigid multi-level registration using the SSD distance measure. The method uses a Gauss-Newton optimization scheme to solve the optimization problem.

**(3) Deformable Registration**    The final step is the matrix-free deformable registration of (König et al., 2018). The deformation is defined as a minimizer of the cost function

$$\min_y \mathcal{D}^{\mathrm{NGF}}(\mathcal{F}, \mathcal{M}(y)) + \alpha \mathcal{R}^{curv}(y), \tag{1}$$

with the normalized gradient field distance measure $\mathcal{D}^{\mathrm{NGF}}$ (Haber and Modersitzki, 2006) that focuses on the alignment of image gradients of the fixed image $\mathcal{F}$ and the deformed moving image $\mathcal{M}(y)$. The second-order curvature regularizer $\mathcal{R}^{curv}$ (Fischer and Modersitzki, 2003) enforces smooth deformation by penalizing spatial derivatives. The parameter $\alpha$ is a weighting factor. The method uses the limited-memory Broyden-Fletcher-Goldfarb-Shannon (L-BFGS) optimization scheme to solve the optimization problem and is embedded in a multi-level scheme.

### 2.3. Lesion Tracking

We use the registration to propagate the baseline contour to the follow-up scan. While this propagated contour may not be accurate enough due to size changes under therapy, it provides a good initial correspondence. To compensate for registration errors, we enlarge the search region by 50 mm in every direction to ensure that the corresponding lesion is inside this selected region and to include enough information for the CNN.

### 2.4. Lesion Selection

The CNN is not constrained to segment only one lesion inside the selected region in the follow-up scan. Therefore, we select the lesion whose center is closest to the center of the propagated lesion. To avoid annotation of wrong lesions close by in the case of vanishing

lesions under therapy, we only accept a lesion annotated by the network if the Euclidean distance of its center is smaller than 25 mm to the propagated lesion center.

## 3. Experiments and Results

### 3.1. Dataset

The dataset consists of 206 baseline and follow-up CT scan pairs of patients with metastatic melanoma (Stage IV, AJCC) treated at the Center for Dermato-Oncology at the University Hospital Tuebingen, Germany. All patients received either mono (Nivolumab or Pembrolizumab) or combined (Nivolumab+Ipilimumab) immunotherapy or targeted therapy (Vemurafenib +Cobimetinib or Dabrafenib+Trametinib) before the follow-up scan. The patients were split into 163 training and validation cases and 43 test cases with overall 2408 and 125 manual annotated soft-tissue lesions in the baseline images. Training was performed exclusively on baseline images, whereas testing was done on both baseline and follow-up scans. Therefore, we selected patients with lower lesion counts for the test set in order to obtain a diverse set of lesions while keeping the annotation effort feasible. For the test cases, 25 of the 125 lesions are gone in the follow-up image.

### 3.2. Baseline Segmentation

To show the advantage of training the network only on a small region of interest around the lesions, we compare our approach to a network trained on the whole images. However, for the evaluation, we use the closest lesion to the point annotation for both approaches, and therefore, false-positive annotations are not taken into account.

Since the network is not forced to segment anything in the region of interest, we evaluate the percentage of correctly annotated lesions. A lesion counts as correctly annotated if there is an overlap with the segmentation mask. To evaluate the performance of our segmentation network, we use Dice coefficient, average surface distance (ASD), and Hausdorff distance (HD) if the network segmented the correct lesion. Moreover, we evaluate the Surface Dice (Nikolov et al., 2018) with a threshold of 1 mm, which is a good approximation for the correction effort given an imperfect segmentation mask of a relatively small structure.

When the nnU-Net is trained only on the small region of interest around the point annotation, the network segments the correct lesion in 96%, whereas with training on the whole image, only 37.6% of the lesions are annotated. On the correctly annotated lesions, the network trained on the ROI achieves on average a better Dice Score (0.79 vs. 0.60), Surface Dice (0.88 vs. 0.68), and average surface distance (1.40 mm vs. 1.77 mm) but a slightly worse Hausdorff distance (5.09 mm vs. 4.59 mm). Note that the number of included lesions for the calculation differs depending on the training mode. Taking all lesions into account the advantage increases to 0.76 vs. 0.23 for the Dice Score and 0.85 vs. 0.26 for the Surface Dice. Figure 2 summarizes the quantitative results and Fig. 3 shows several visual examples of the results produced by our network.

### 3.3. Registration Accuracy

We measure the registration accuracy using the center point matching (CPM) accuracy as in (Cai et al., 2020), which represents the percentage of correctly matched lesions. A match

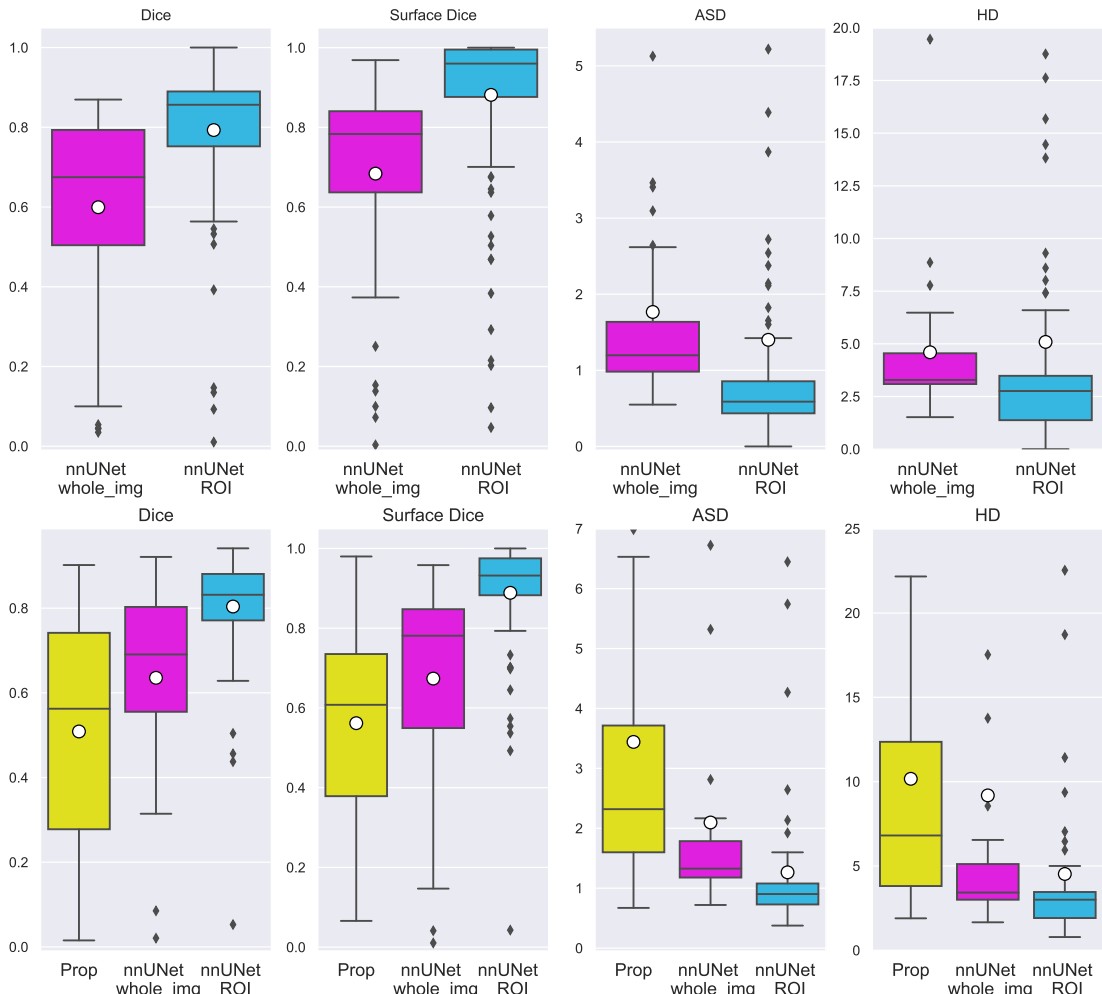

Figure 2: Comparison of the evaluation metrics for all baseline lesions (upper row) and follow-up lesions (lower row) in which the correct lesion was annotated with the underlying training mode. Therefore, the number of included lesions for the calculations varies depending on the training mode. For the follow-up lesions, the lesion results by the registration propagated are shown in yellow. The boxplots show the median line and the mean as a white circle.

counts as correct when the Euclidean distance between the center of the propagated baseline lesion and the center of the manually annotated follow-up lesion is smaller than a threshold. Since in this application whole-body CT scans are registered and large volume changes of the lesion happen due to therapy, we set the threshold to 25 mm. For this evaluation, only the lesions which are visible in the follow-up image are taken into account and therefore the number of lesions reduces to 100.

In 95 of the 100 cases, the Euclidean distance was less than the threshold with a mean Euclidean distance of 7.66 mm. The average absolute offset between the center of the

propagated baseline lesion and the center of the manually annotated follow-up lesion is 3.79 mm, 3.16 mm and 4.49 mm in x-, y- and z-direction, respectively. A histogram of the offset is shown in Fig. 8 in the appendix.

### 3.4. Follow-Up Segmentation

We evaluate the follow-up segmentation in the same way as the baseline segmentation. However, the successful segmentation of the follow-up lesion depends not only on the segmentation accuracy itself but the whole pipeline. For the cases in which the lesion was not propagated accurately enough, segmentation by the nnU-Net was not possible. To evaluate the whole pipeline, those lesions are counted as not correctly annotated lesions. Furthermore, in the 25 cases in which the lesion was fully regressive in the follow-up image, we expect the nnU-Net not to annotate anything.

In 80% of the lesions, our pipeline successfully annotates the lesion in the follow-up scan with an average Dice Score of 0.80 and an average Surface Dice of 0.89. The lesion propagated by the registration has an overlap to the manual annotation in 77.5% with an average Dice score of 0.51 and a Surface Dice of 0.56. All quantitative results are summarized in Fig. 2. All failure cases are visualized in the appendix. In 17 of the 25 cases in which the lesion has disappeared in the follow-up image, the nnU-Net correctly not segment anything.

## 4. Discussion and Conclusion

This paper presents a pipeline that automates the segmentation of matching lesions in follow-up CT examinations of cancer patients, given a one-click point annotation in the baseline lesion. We have validated our pipeline on the challenging task of whole-body soft-tissue lesion tracking and segmentation. Our pipeline succeeded for 96% of the baseline lesions and for 80% of the follow-up lesions with an average Dice Score of 0.79 and 0.80, respectively. Furthermore, our pipeline achieves an average Surface dice of 0.88, which shows that the required correction effort is very low.

All failure cases in the follow-up image are visualized in Fig. 6 in the appendix showing that the pipeline fails due to different reasons. For some cases, the registration was not accurate enough and therefore a wrong or no lesion was selected even though the correct one was segmented. Other lesions are hard to distinguish from surrounding tissue or they have an untypical shape that might cause problems. In some cases, the lesion split into two smaller lesions in the follow-up scan after the patient received therapy and the nnU-Net segmented both, but just one lesion was selected. In some of these cases, it is also difficult for a radiologist to identify and segment the lesion correctly. Our pipeline has still some limitations which have to be addressed before it could be used in the clinic. Lesions can split or merge over time, however, our pipeline assumes that every lesion in the baseline has zero or one corresponding lesion in the follow-up image. This does not always have to be true. Moreover, lesions that are very close to each other could be wrongly assigned in the follow-up scan. These problems will be solved in future work by integrating consistency rules. Besides, our pipeline is not yet capable of detecting new lesions in the follow-up scan. Furthermore, the current pipeline does not take the appearance of the baseline lesion into account. There are different approaches to integrate this information into the model. The transformed baseline image and the corresponding lesion mask could be used as an additional input for

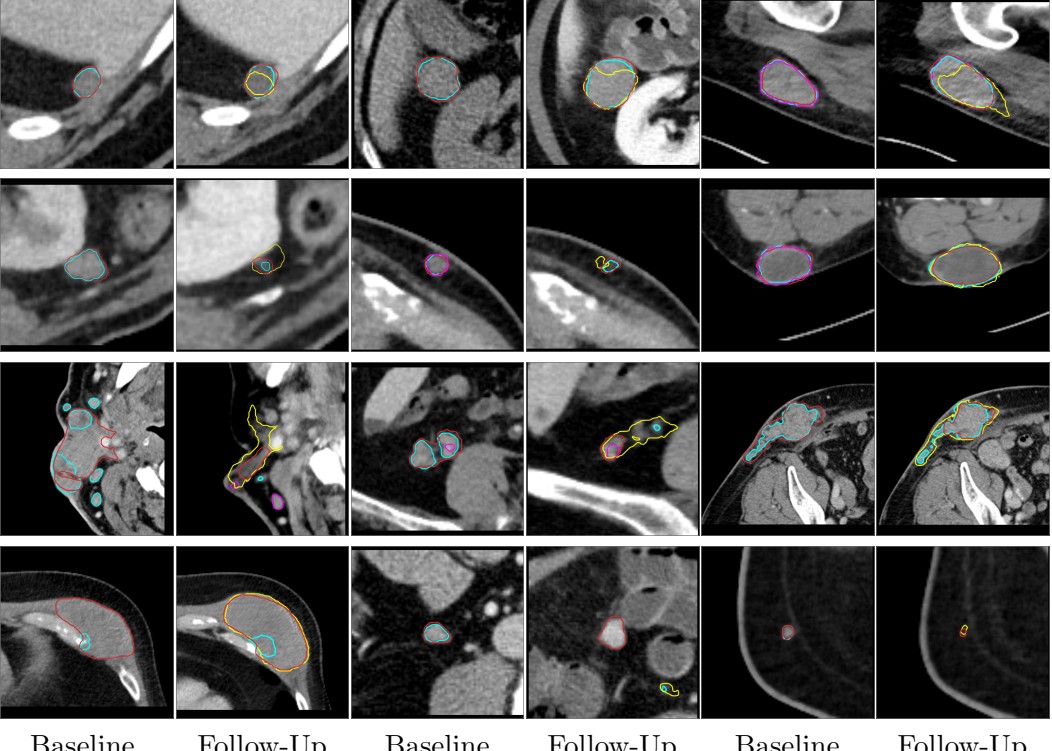

Baseline    Follow-Up    Baseline    Follow-Up    Baseline    Follow-Up

Figure 3: Visual examples of results produced by our method. Each example includes the baseline and follow-up lesion and therefore consists of two images (left baseline, right follow-up). On the baseline image, the manual annotation (red curve ■) and the nnU-Net annotation trained on the ROI (blue curve ■) and trained on the whole image (pink curve ■) are shown. On the follow-up image, the manual annotation (red curve ■), the propagated lesion (yellow curve ■) and the results of the presented pipeline (blue curve ■) are visualized.

the follow-up model. However, this would mean that two models have to be trained; one for segmenting the baseline image and one for the follow-up images. To train the follow-up network, a sufficient number of lesion annotations has to be available. Unfortunately, we only have the annotations that we used for the evaluations and therefore this approach is not suitable. Another approach is a joint-segmentation-registration algorithm as in (Li et al., 2019). We will explore this approach in future work.

We have trained and evaluated our method on soft-tissue lesions, which are particularly challenging due to their diverse appearance and location. Our promising results suggest that we will be able to extend our approach to other lesion types as well. Additionally, for use in clinical routine, it is sufficient to extract the largest diameter from the segmentation, so that detailed corrections will not be necessary. With our work, we have laid the foundation for an efficient automated follow-up assessment according to the RECIST standard and implementation of automated segmentation for Radiomics analysis in clinical routine.

## Acknowledgments

We thank Fabian Isensee, Paul Jäger, Simon Kohl, Jens Petersen, and Klaus Maier-Hein for providing the nnU-Net framework. The research was funded by the Deutsche Forschungs-gemeinschaft (DFG, German Research Foundation) – 428216905 / SPP 2177.

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

## Appendix A. Dataset

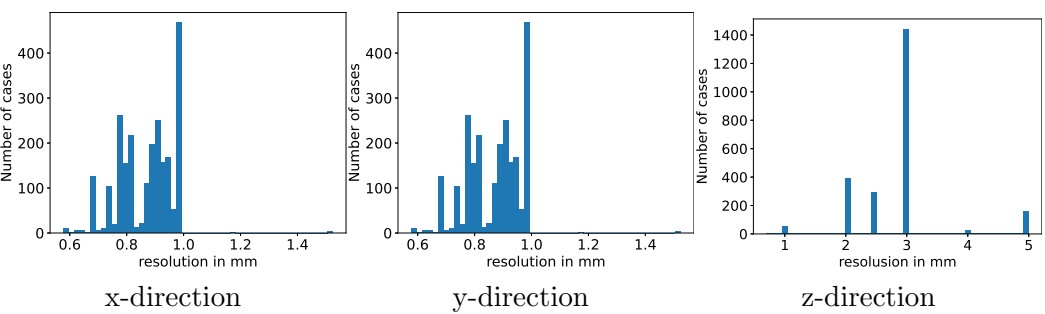

Figure 4: Histogram of the image resolution in x-,y- and z-direction.

## Appendix B. Visualization of Deformationfields

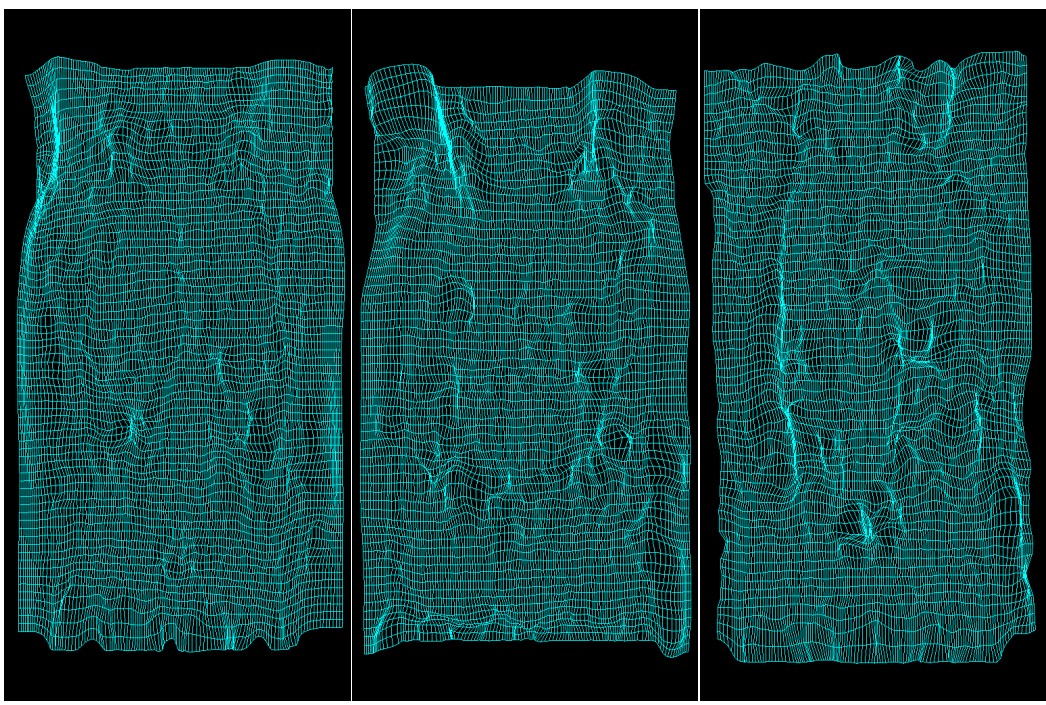

Figure 5: Example coronal slices extracted from three deformation fields to give an impression of the smoothness.

## Appendix C. nnUNet settings

Table 1: Main settings chosen by the nnUNet framework to train the segmentation network

| name | description | chosen parameter |
|---|---|---|
| net_pool_per_axis | number of pooling operations in z,x,y direction | 3,5,5 |
| base_num_features | number of features after first conv | 32 |
| conv_per_stage | | 2 |
| optimizer | | SGD |
| learning rate | | $\approx 0.00235$ |
| max_num_epochs | maximal number of epochs | 1000 |
| num_batches_per_epoch | number of batches used in every epoch | 250 |
| batch_size | number of images per batch | 5 |
| patch_size | z,y,z direction | 56 128 128 |
| normalization_schemes | see (Isensee et al., 2020) for details on CT scheme | (0,'CT') |

## Appendix D. Failure cases

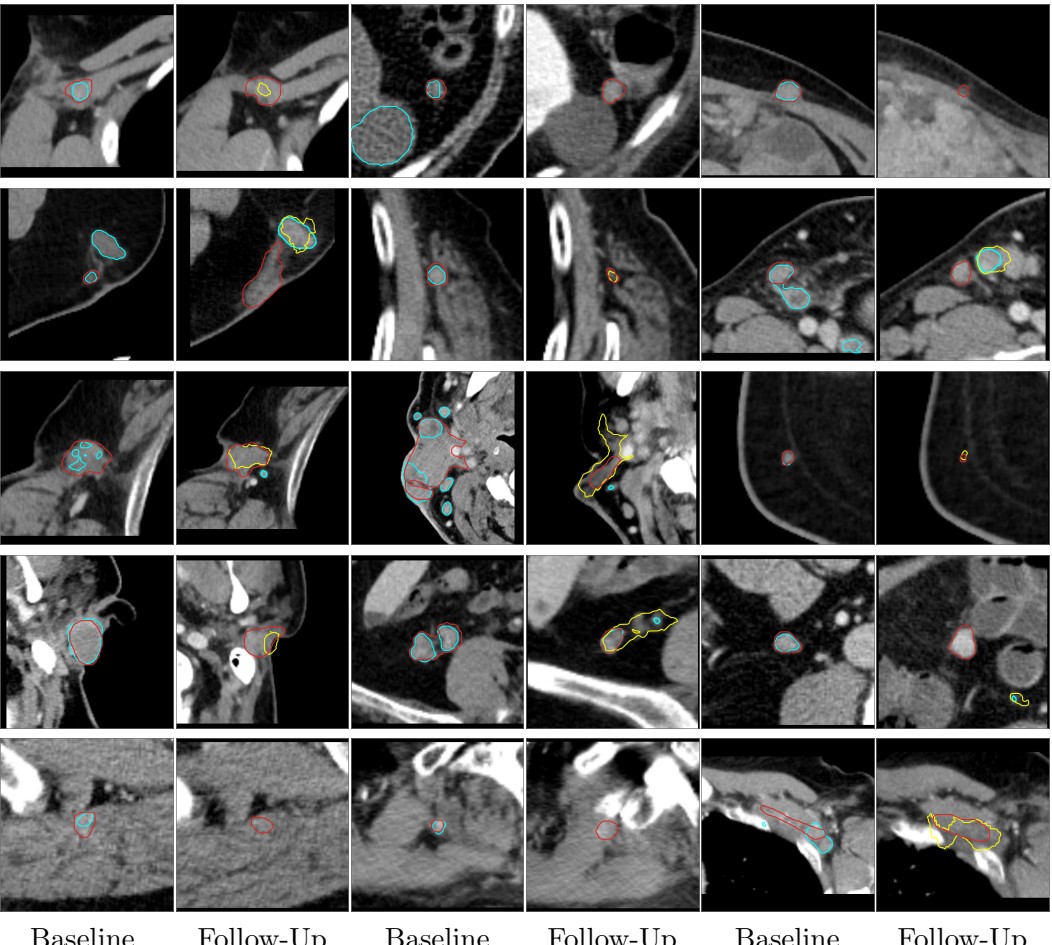

| Baseline | Follow-Up | Baseline | Follow-Up | Baseline | Follow-Up |

Figure 6: All cases in which our pipeline fails to segment the lesion in the follow-up image. Each example includes the baseline and follow-up lesion and therefore consists of two images (left baseline, right follow-up). On the baseline image, the manual annotation (red curve ■) and the nnU-Net annotation trained on the ROI (blue curve ■) are shown. On the follow-up image, the manual annotation (red curve ■), the propagated lesion (yellow curve ■) and the results of the presented pipeline (blue curve ■) are visualized. For these cases, we do not apply the lesion selection and therefore some lesions seem to be correctly segmented, however, they are not selected using our criteria. There are different reasons for these failures. In some cases, the registration was not accurate enough and therefore a wrong or no lesion was segmented. Some lesions are hard to distinguish from surrounding tissue (e.g. last column), but also an untypical shape can be a problem.

## Appendix E. Diameter Error

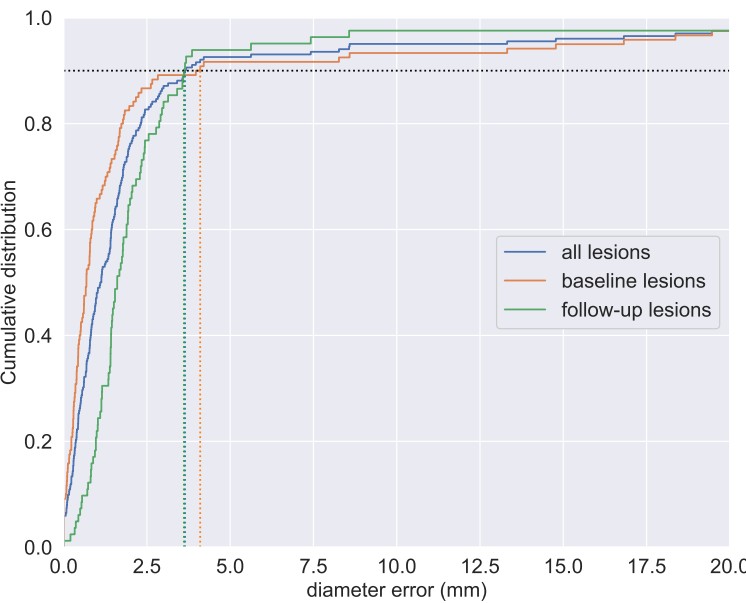

Figure 7: Cumulative distribution of diameter error between the manual segmentation and the nnU-Net segmentation. Please note, that in clinical routine the diameter would not be calculated from a segmentation but measured directly which might also introduce some errors. The dotted lines visualize the 90th percentiles of the error, which are 3.6 mm for all lesions, 4.1 mm for the baseline lesions and 3.6 mm for the follow-up lesions.

## Appendix F. Registration Accuracy

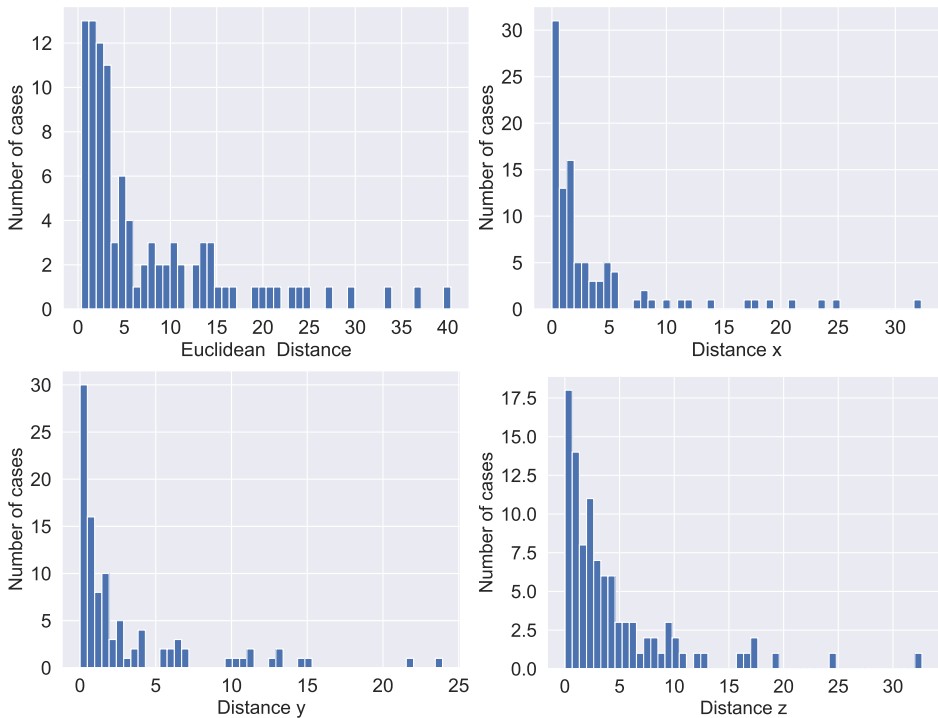

Figure 8: Histogram of Euclidean distance and the absolute offset between the center of the propagated lesion and the center of the manually annotated follow-up lesion..

