# OpenReview forum: "Whole-Body Soft-Tissue Lesion Tracking and Segmentation in Longitudinal CT Imaging Studies"
_MIDL.io/2021/Conference — MIDL 2021_

### Official Review · AnonReviewer1 · 2021-03-07

**Confidence:** 4
**Preliminary Rating:** 3
**Recommendation:** Poster
**Final Rating:** 4

**Summary:**

A pipeline to reach consistently segmented lesions across longitudinal CT scans, by propagating the segmentation of baseline images (achieved with nnU-net) using the transformation between baseline and follow-up images (achieved by registration), and using its output to define a segmentation ROI and/or to select the segmentation of follow-up images that corresponds to the propagated lesion. Evaluation on 125 lesions from 43 patients, with 80% of correctly annotated follow-up lesions.

**Strengths:**

- A relevant combination of segmentation and tracking to reach consistency in challenging follow-up segmentations
- Demonstrated improvements in performance on a reasonable database
- Clear writing
- Valuable reporting of failing cases

**Weaknesses:**

- A rather applicative framework that revisits standard segmentation+registration schemes with state-of-the-art methods
- No use of the transformation from baseline to follow-up to characterize lesion changes

**Deanonymize Review:**

no

**Detailed Comments:**

- The Abstract may explain what is meant by “success”
- The Introduction could develop a bit more on segmentation across longitudinal scans not necessarily in CT, in particular as the combination of segmentation + registration was topical for such problems before the advent of deep learning.
- Sec.2.2: the authors may detail a bit what is behind the “normalized gradient field distance measure” and why this metric may be interesting for these images
- Writing issues:
_ Sec.2.1: “to train 3d” > “to train a 3d”
_ Sec.2.1: “in the baseline and follow-up image” > “… images”

**Final Rating Justification:**

I appreciate the answers made to my comments and those from the other Reviewers, leading to necessary (slight) updates.
I've updated my rating, which now agrees with the other Reviewers.

**Justification Of The Preliminary Rating:**

A rather applicative framework that should be better situated vs. previous segmentation+registration approaches, still with a relevant strategy to reduce errors in follow-up segmentation, but tracking could be better exploited beyond the segmentation purpose.

**Paper Type:**

validation/application paper

**Questions To Address In The Rebuttal:**

- Are lesion changes from baseline to follow-up a valuable information? If so, would exploiting the deformation field from registration be of value? Could the authors develop at least a bit on this? If not exploited, this should be mentioned in the future work p.8.
- Could they also provide 1-2 deformation fields to visualize their smoothness?

**Special Issue:**

no

---

> ### Author Response · Authors · 2021-03-16
>
> We thank you for the time you took to review thoroughly our paper. We are very pleased with the kind and constructive comments we received.
>
> * "Are lesion changes from baseline to follow-up a valuable information? If so, would exploiting the deformation field from registration be of value? Could the authors develop at least a bit on this? If not exploited, this should be mentioned in the future work p.8."
>
>   This is an excellent question. We also assume that it is a helpful information. During our experiments, we already considered different approaches to integrate this information into our model:
>
>   1. Use the transformed baseline image and the corresponding lesion mask as additional inputs for the follow-up model. However, this would mean that we need two different CNNs – one for segmenting the baseline image and one for the follow-up image. To train the follow-up network, we would need sufficient lesion annotation on follow-up images. Unfortunately, we currently only have the annotations that we used for the evaluation. Therefore, we have not yet been able to further investigate this approach.
>   2. Use the deformation field as additional input (+ information from 1.). We have the same problems as described in 1.  Furthermore, we should consider the possibility that this information is harder to process for a segmentation network.
>   3. Directly learn a joint-registration-segmentation network (on the whole image). Without using a conventional registration method in between, we have to use the whole image learning such a network. However, due to different field of views and large images sizes (up 1000 slices), we probably get memory problems or have to downsample the image too much so that there is not enough image information left for an accurate registration.
>   4. First perform a conventional registration method to obtain corresponding regions and then learn a joint-segmentation-registration network based on the images patches. We still have to consider how we handle the problem of missing follow-up annotations but we have some ideas to solve this problem. However, we couldn’t use the nnUNet framework for this approach anymore rather building a completely new method. We are working on developing deep-learning-based registration methods as well (e.g. [1,2]) but adding a novel joint-registration-segmentation approach into this paper would exceeds the scope of this work.
>
>  To summarize, we have thought about this approach but will further investigate it in a future journal paper. Our radiologist will further annotate the follow-up lesions so that different approaches can be compared. We will add a short outlook of the above described experiments in the discussion section.
>
>  [1] Hering, A., van Ginneken, B., & Heldmann, S. (2019, October). mlvirnet: Multilevel variational image registration network. In International Conference on Medical Image Computing and Computer-Assisted Intervention (pp. 257-265). Springer, Cham.
>  [2] Hering, A., Häger, S., Moltz, J., Lessmann, N., Heldmann, S., & van Ginneken, B. (2020). Constraining Volume Change in Learned Image Registration for Lung CTs. arXiv preprint arXiv:2011.14372.
>
>
>
> * "Could they also provide 1-2 deformation fields to visualize their smoothness?"
>
>  Yes, we will add those in the appendix.
>
> * "The Introduction could develop a bit more on segmentation across longitudinal scans not necessarily in CT, in particular as the combination of segmentation + registration was topical for such problems before the advent of deep learning."
>
>  Do you have some specific papers of conventional segmentation+registration paper in mind? Maybe there are some works we don’t know. We will try to add a bit on this topic to our introduction. However, since we cannot further extend our paper due to the page limit, we probably have to keep it short to have space left for a more detailed discussion.

---

### Official Review · AnonReviewer3 · 2021-03-08

**Confidence:** 5
**Preliminary Rating:** 4
**Recommendation:** Oral

**Summary:**

The authors present an image processing pipeline to segment soft tissue tumors in longitudinal CT scans. The user clicks in one lesion at the first CT scan. The proposed method segments it, register the subsequent CT scan to the original one, propagates the segmentation and corrects it on the target image. The dataset used to validate the method consists in 206 baseline and follow up CT scan pairs, split in 163 training and validation and 43 test cases. There are 125 manual annotated lesions on the test set. Lesion segmentation results are good, with a DICE score of 0.8. 80% of the lesions were successfully annotated on the follow-up scan.

**Strengths:**

-	Interesting application
-	Good pipeline design and implementation
-	Good results
-	Very clear exposition
---------------------------------------------------------------------------------------------------

**Weaknesses:**

-	It is not clear how the nnUnet is trained. More details would be appreciated.
-	There is no discussion on the appearance of new lesions. If we were to completely automate the problem of assessment of treatment response, we would need to have methods to detect new lesions, which is a biomarker of (very) poor prognosis. The authors acknowledge such problem when lesions split, but it can also happen that new metastases appear at distant locations or organs.


**Deanonymize Review:**

no

**Justification Of The Preliminary Rating:**

The topic is interesting. The pipeline is well defined. The results are compelling. Minor improvements can be done to the paper, such as the ones mentioned on the weaknesses.
--------------------------------

**Paper Type:**

both

**Special Issue:**

yes

---

> ### Author Response · Authors · 2021-03-16
>
> We thank you for the time you took to review thoroughly our paper. We are very pleased with the kind and constructive comments we received.
>
> * "It is not clear how the nnUnet is trained. More details would be appreciated. "
>   * Thank you for pointing that out. We added a section in the appendix with the most important parameters of the nnUNet.
> * "There is no discussion on the appearance of new lesions. If we were to completely automate the problem of assessment of treatment response, we would need to have methods to detect new lesions, which is a biomarker of (very) poor prognosis. The authors acknowledge such a problem when lesions split, but it can also happen that new metastases appear at distant locations or organs."
>   * We agree with you that this is a highly important problem regarding completely automating the assessment of treatment response. As we described briefly in our paper, the one-click solution in the baseline image was introduced to avoid false positive lesions. The same applies initially to the follow-up image. However, as you correctly mentioned, the appearance of new lesions is a biomarker of poor prognosis and therefore new lesions have to be detected. We added a sentence in our discussion that makes clear that our pipeline is not yet capable of detecting new lesions. However, the main goal of our approach is not to fully automate the oncological assessment but assist the radiologist in reading those scans. That means that we speed up the process by precomputing as much as possible by automatically segmenting and measuring the selected lesions. For new lesions in the follow-up scan, the radiologist can use the one-click approach to measure them. Nevertheless, we would like to address that issue in future work.

---

### Official Review · ~Vincent_Jaouen1 · 2021-03-10

**Confidence:** 5
**Preliminary Rating:** 4
**Recommendation:** Poster
**Final Rating:** 4

**Summary:**

This paper proposes a nearly automatic pipeline for the segmentation of soft tissue lesions (STL) in longitudinal whole body CT acquisitions. Manual interaction is reduced to the selection of the location of center of the STL in the baseline (BL) CT scan. During training a 3d nnU-Net is trained in bounding boxes of 100x100x100mm3 surrounding the STL in BL scans.
During deployment, the STL in BL is first segmented. Then the lesion mask is propagated between the BL and longitudinal scan using a 3-stage registration approach (translation, then rigid, then elastic).  The search space in is then extended  before segmentation inference on the followup scan.


**Strengths:**

- The authors propose a sound methodology for an important practical clinical issue of tracking/follow up of lesions in longitudinal imaging.
- the paper is very well written
- experiments are performed on a large cohort of 206 baseline-follow up pair
- results are very convincing with a strong improvement over a whole body nnUnet
- nice description of failure cases

**Weaknesses:**

- no major methodological contribution but a sound pipeline
- the inference on the follow up scan is not guided by the BL mask, which is a limitation that should be more discussed and could likely address some of the failure cases mentioned in the paper

- one can regret such an unbalanced split between train and test (ntrain=163 vs ntest=43) given the large size of the cohort. nnUnet is normally robust to small cohorts with extensive 5 fold cross validation training. I understand these are whole body scans and the network may require such a number of images to perform well. If this is the reason, this could have been evaluated.


**Deanonymize Review:**

yes

**Detailed Comments:**

- It would have been nice to mention further developments including e.g. strategies taking into account the initial segmentation mask for the follow up image.

**Final Rating Justification:**

The authors have addressed all my questions by providing very detailed answers and modifications to the paper when needed. I therefore keep my strong accept rating.

**Justification Of The Preliminary Rating:**

The paper addresses a practical problem in a very principled multi-stage fashion. While no clear methodological novelties are proposed, these guidelines are very likely to be applicable with few modifications to other modalities and pathologies. Results are convincing and validated on a large cohort. I therefore strongly recommend acceptation.


**Paper Type:**

methodological development

**Questions To Address In The Rebuttal:**

- In the discussion  "For some cases, the registration was not accurate enough and therefore a wrong or no lesion was selected even though the correct one was segmented". This needs to be further characterized. For example,  it seems there is a bias towards error along the z axis with quite a number of >5mm cases. Is this due to slice thickness ? Are z errors due to the translational or the rigid stage ?  Could the translational registration be improved to better locate the z center. Could you think of allowing only rigid aligment in a plane ?

- please specify the resolution/thickness (+-std) of the CT scans + additional info such as vendor/model if mono center trial.

- please provide also histograms of Euclidean distance in Fig. 6 appendix C to better characterize distribution tails.

 - Did the authors try a smaller training set ? It would have been nice to see the performance of the pipeline as a function of the number of training images

- please specify how the difference in size between train and test search space is accommodated. Does nnUnet natively handle inference on different images sizes ?

**Special Issue:**

no

---

> ### Author Response · Authors · 2021-03-16
>
> We thank you for the time you took to review thoroughly our paper. We are very pleased with the kind and constructive comments we received.
>
> * "the inference on the follow up scan is not guided by the BL mask, which is a limitation that should be more discussed and could likely address some of the failure cases mentioned in the paper"
>
>
>   We agree with you that this information might be helpful to achieve better results. During our experiments, we already considered different approaches to integrating this information into our model:
>
>  1. Use the transformed baseline image and the corresponding lesion mask as additional inputs for the follow-up model. However, this would mean that we need two different CNNs – one for segmenting the baseline image and one for the follow-up image. To train the follow-up network, we would need sufficient lesion annotation on follow-up images. Unfortunately, we currently only have the annotations that we used for the evaluation. Therefore, we have not yet been able to further investigate this approach.
>  2. Use the deformation field as additional input (+ information from 1.). We have the same problems as described in 1.  Furthermore, we should consider the possibility that this information is harder to process for a segmentation network.
>  3. Directly learn a joint-registration-segmentation network (on the whole image). Without using a conventional registration method in between, we have to use the whole image learning such a network. However, due to different field of views and large images sizes (up 1000 slices), we probably get memory problems or have to downsample the image too much so that there is not enough image information left for an accurate registration.
>  4. First perform a conventional registration method to obtain corresponding regions and then learn a joint-segmentation-registration network based on the images patches. We still have to consider how we handle the problem of missing follow-up annotations but we have some ideas to solve this problem. However, we couldn’t use the nnUNet framework for this approach anymore rather building a completely new method. We are working on developing deep-learning-based registration methods as well (e.g. [1,2]) but adding a novel joint-registration-segmentation approach into this paper would exceeds the scope of this work.
>
>  To summarize, we have thought about this approach but will further investigate it in a future journal paper. Our radiologist will further annotate the follow-up lesions so that different approaches can be compared. We will add a short outlook of the above-described experiments in the discussion section.
>
>  [1] Hering, A., van Ginneken, B., & Heldmann, S. (2019, October). mlvirnet: Multilevel variational image registration network. In International Conference on Medical Image Computing and Computer-Assisted Intervention (pp. 257-265). Springer, Cham.
>  [2] Hering, A., Häger, S., Moltz, J., Lessmann, N., Heldmann, S., & van Ginneken, B. (2020). Constraining Volume Change in Learned Image Registration for Lung CTs. arXiv preprint arXiv:2011.14372.
>
>
> * "one can regret such an unbalanced split between train and test (ntrain=163 vs ntest=43) given the large size of the cohort. nnUnet is normally robust to small cohorts with extensive 5 fold cross validation training. I understand these are whole body scans and the network may require such a number of images to perform well. If this is the reason, this could have been evaluated."
>
>  The chosen split might not see optimal but due to the high variety of lesions (locations, shapes, etc.) a large number of training cases is beneficial. Moreover, it had practical reasons. All baseline lesions were annotated first for research in the context of radiomics. To perform the presented research, the radiologist annotated a few follow-up scans but not all. And as we mentioned in the paper, we selected patients with lower lesion counts for the test set in order to obtain a diverse set of lesions while keeping the annotation effort feasible. Since we want to use the same network and the same patient split for the baseline and follow-up lesions, we had to take the drawback of a non-optimal split. However, our radiologist will annotate all missing follow-up lesions using the presented pipeline. Therefore, we predict the follow-up lesion and he will correct them. Moreover, he will rate the quality of the automatically computed lesion to have a further accuracy measure besides the one already used.

---

> > ### Author Response · Authors · 2021-03-16
> >
> > * "Did the authors try a smaller training set? It would have been nice to see the performance of the pipeline as a function of the number of training images"
> >
> > We did not try a smaller training set. As described above, we split our dataset into training and test depending on the presence of the follow-up lesion annotation. Subsequently, we used all baseline lesions of the training set to train the nnUNet to achieve the best possible results. Nevertheless, we agree with you that this might be an interesting topic to investigate.
> >
> > * "please specify the resolution/thickness (+-std) of the CT scans + additional info such as vendor/model if mono center trial. "
> >
> >
> >  We added this information in the dataset section. Furthermore, we added a histogram of the image resolution in the appendix. The images have a mean resolution of 0.866+-0.091 in plane and a mean thickness of 2.86+-0.74.
> > In 22 training and 2 test cases, the baseline CT scan was acquired externally and therefore no vendor information is available. For all internal baseline scans, one of the following scanners were used: Siemens Somatom Defeniton AS, Siemens Somatom Defeniton Flash und Siemens Somatom Force. We would assume that due to the external cases during training, our network might be more robust concerning scanner differences. Unfortunately, the number of external cases is not high enough to draw any conclusion on that.
> >
> > * "please provide also histograms of Euclidean distance in Fig. 6 appendix C to better characterize distribution tails."
> >
> >
> > Thanks for this suggestion. We added this to the appendix.
> >
> > * "please specify how the difference in size between train and test search space is accommodated. Does nnUnet natively handle inference on different images sizes ?"
> >
> >
> >   Yes, the nnUNet handles this. It uses a sliding window approach with fixed patch size for inference.
> >
> > * " In the discussion "For some cases, the registration was not accurate enough and therefore a wrong or no lesion was selected even though the correct one was segmented". This needs to be further characterized. For example, it seems there is a bias towards error along the z axis with quite a number of >5mm cases. Is this due to slice thickness ? Are z errors due to the translational or the rigid stage ? Could the translational registration be improved to better locate the z center. Could you think of allowing only rigid aligment in a plane ?"
> >
> >
> >   Since we discard a lesion if the distance between the center of the propagated lesion and the segmented one (by the nnUNet) is more than 25mm, we also discard lesions if the registration was not accurate enough. The reason for this is that we want to prevent wrong annotations if a lesion has vanished due to therapy. We have thus opted for a compromise between the two desired characteristics.
> > But yes, the main reason for the registration inaccuracy in the z-direction is the slice thickness. Most of the images have a thickness of 3mm. We will further investigate if we can improve the rigid alignment so that it will give a better starting point for the deformable registration. However, we are not sure if restricting the transformation to a rigid alignment inplane will reduce the registration error. There are many non-rigid motions inplane like breathing or weight loss during therapy.

---

### Meta-Review · Area_Chair1 · 2021-03-29

**Recommendation:** Accept (Oral)

**Metareview:**

Longitudinal lesion tracking is a very clinical relevant problem that has received little attention in deep learning. This paper proposes an interesting semi-automatic pipeline with a  one-click interaction combining segmentation and registration to find the lesions in the follow-up exam. The method builds on existing segmentation (nnunet) and registration bricks but the conceptual connection, the one-click interaction, and the medical application are sound and principled. Clarity and discussion on failure cases were praised by R1 and R3. R1 suggests a deeper discussion of the prior work on joint registration and segmentation outside of the application. An issue raised by R2 and R1 is the fact that the follow-up does not take into account the initial segmentation mask. The authors provide in the rebuttal some alternatives and discuss the practical reasons preventing the implementation of such solutions. A similar discussion should be added to the paper.

**Paper Type:**

validation/application paper

---

### Decision · Program_Chairs · 2021-03-31

**Decision:**

Accept

**Comment:**

Congratulations your paper has been selected as long oral.